# Cell-Free Supernatant (CFS) from *Bacillus subtilis* EB2004S and *Lactobacillus helveticus* EL2006H Cultured at a Range of pH Levels Modulates Potato Plant Growth under Greenhouse Conditions

**DOI:** 10.3390/ijms24076620

**Published:** 2023-04-01

**Authors:** Levini A. Msimbira, Judith Naamala, Sowmyalakshmi Subramanian, Donald L. Smith

**Affiliations:** Department of Plant Science, McGill University, Montreal, QC H9X 3V9, Canada; levini.msimbira@mail.mcgill.ca (L.A.M.); judith.naamala@mail.mcgill.ca (J.N.); sowmyalakshmi.subramanian@mcgill.ca (S.S.)

**Keywords:** plant growth-promoting microbes, cell-free supernatant, pH–microbe–plant interaction, biostimulant, soil acidity and alkalinity

## Abstract

Agriculture involving industrial fertilizers is another major human made contributing factor to soil pH variation after natural factors such as soil parent rock, weathering time span, climate, and vegetation. The current study assessed the potential effect of cell-free supernatant (CFS) obtained from *Bacillus subtilis* EB2004S and *Lactobacillus helveticus* EL2006H cultured at three pH levels (5, 7, and 8) on potato (var Goldrush) growth enhancement in a greenhouse pot experiment. The results showed that CFSs obtained from *B. subtilis* EB2004S and *L. helveticus* EL2006H cultured at pH 5 significantly improved photosynthetic rates, stomatal conductance, root fresh weight, and whole plant fresh weight. interactive effects of pot pH and that of CFSs obtained from pH 5 influenced chlorophyll, plant height, and shoot and whole plant fresh weight. Moreover, treatment 52EB2004S~0.4% initiated early tuberization for potato grown at pH 7 and 8. Potato grown at pH 5, which received a 72EB2004S~0.4% CFS treatment, had greater whole plant fresh and dry weight than that treated with *L. helveticus* EL2006H CFS and a positive control. Taken together, the findings of this study are unique in that it probed the effect of CFS produced under differing pH conditions which revealed a new possibility to mitigate stresses in plants.

## 1. Introduction

Soil pH (a measure of the activity of hydrogen ions in soil solution) affects all chemical and geochemical processes in the soil with ultimate impact on the soil nutrient supply, affecting survival and performance of both plants and microbes. Plant growth under natural environmental settings relies on, among other things, pH as a master variable; geospatial variations show strong correlation with plant species richness [1], including variability among soil profile horizons [2,3]. This makes pH a first variable to be checked whenever plants do not perform as expected, whether under field conditions or under controlled environment conditions, such as greenhouses, making it analogous to a patient’s body temperature check during diagnosis by medical doctors [4]. Microbes exert activities largely through soil enzymes, the presence and activities of which are influenced by pH [5,6,7]. Soil pH is additionally important in biodegradation processes in the environment and has shown its importance in the degradation of compounds such as atrazine [8]. The dominant chemical indicator for soil health is generally pH, which is often a function of inherent parent rock, weathering time span, climate, and vegetation [4,9]. In this respect pH influences soil health and, subsequently, crop plant growth and yield; the United States Department of Agriculture (USDA) categorized soil pH as: acidic (ultra < 3.5, extreme 3.5–4.4, very strong 4.5–5.0, strong 5.1–5.5, moderate 5.6–6.0, and slight 6.1–6.5), neutral (6.6–7.3), and alkaline (slight 7.4–7.8, moderate 7.9–8.4, strong 8.5–9.0, and very strong >9.0) (Burt, 2014). This corresponds to the known optimum growth requirements for most terrestrial plants which fall within a range of pH 6.5–8.0 [9]. Therefore, optimum plant growth potential is generally between 6.5 (slightly acid) and 7.5 (slightly alkaline).

Many soils naturally tend to acidify over time due to rainwater, leaching of excess nitrates, applications of fertilizers (urea and anhydrous ammonia), land use, and decomposition of organic matter [3,10], while alkalization results from accumulation of large amounts of carbonates in the soil [11,12]. The pH-related soil challenges have continued to intensify with time as a result of agriculture and other human activities, such as mining. A range of intervention practices have been implemented and their efficacy is evaluated at different capacities, with the most utilized soil acidity amendment practice being liming [13], a process that helps in raising soil pH to revitalize soil for crop production and other soil–plant interactions including nutrient uptake. Liming is a site-specific soil acidity amendment technique which lacks long-term sustainability as it is highly influenced by cost–benefit analysis and economics [13]. An encouraging direction in dealing with various stresses and their effects on plant production is the emergence of ecofriendly activities, in particular, the use of microbes and microbial-related products to confer resilience in the face of multivarious plant stress challenges, including acidity and alkalinity [12,14].

Potato is currently the fourth most important food crop in the world after wheat, rice, and corn [15,16]. Shifts in food preferences and affordability influence increased production of one crop over another. Potato is among the crops whose consumption has steadily increased in recent years, replacing other crops which were previously dominant [17]. The projected increase in consumption of potato in the future demands sustainable means of production to avoid environmental decline. North American farmers, including Canadians, for instance, apply higher doses of synthetic fertilizers, especially phosphorus (P), to ensure potato nutrient requirements are met [18,19]. The fact that potato is low in P use efficiency has led to build-up of P levels in soils [20]. The use of microbes or microbial products to facilitate availability of the 70 to 90% of P already available in soils [21], but unavailable to plants, is one of the ways which would help reduce the build-up of P in the soils. However, compound fertilizers are known to lower soil pH, leading to diminished soil health. It is therefore imperative to use plant growth-promoting microbes (PGPMs) [22] and PGPM-based products [23,24] along with other techniques, such as biochar, mulching, and rotational cropping [25,26].

The association of plants and microbes is clear in both positive and negative effects. Plant growth-promoting microbes (PGPMs) use various mechanisms for plant growth promotion which include bioactive compound (phytohormones, peptides, and volatile compounds) production, mineral solubilization, N_2_ fixation, enhanced nutrient use [27], and production of growth-stimulating signal compounds [28]. Mechanisms related to antagonistic behavior and/or antibiosis also indirectly enhance plant growth and stress tolerance [28,29]. Given the current understanding of these ecosystem services, the full force of the considerable potential benefits is yet to be determined. It is known that the efficacy of PGPM inoculation depends on soil pH, ability of the microbe to compete with the native strains, temperature, and host specificity [30]. Microbial strains, therefore, contribute to alleviating effects of stresses on plants such as nitrogen deficiency [31], drought [32,33], salinity [34,35,36], acidity, and alkalinity [12,37]. However, the use of microbes or microbial products in dealing with acidity and alkalinity stress in plants is still fragmented. The current study evaluating two plant growth-promoting strains, *Bacillus subtilis* EB2004S and *Lactobacillus helveticus* EL2006H previously evaluated by Msimbira et al. [38], is among the very few studies of this nature. The study has specifically focused on understanding the effect of using CFSs of the strains on potato, cultured at a range of pH levels, rather than using live cells of the microbes. This study was motivated by the fact that live microbes suffer multiple constraints when applied directly, including competition with microbes already in the soil and other environmental stresses [39], pH included. In this study, potato growth performance was assessed specifically in response to pH and CFSs obtained from *B. subtilis* EB2004S and *L. helveticus* EL2006H, two of the principal strains of a microbial consortium produced by EVL Inc. (http://www.evlbiotec.com, accessed on 10 February 2023), a Canadian company.

## 2. Results

### 2.1. The Effect of pH and CFS Treatments on Potato Growth and Physiological Variables

Leaf greenness indices are important both in understanding growth and overall performance of plants. Potato plants’ leaf area (LA) and leaf greenness (LG) in this study indicated dependence on pH and CFS treatments. The results from the two-way ANOVA revealed significant effects of pH on LG for both experiments in the first 30 DAP (Table 1). The LG index was significantly higher at pH 5, measured as 45.89 and 50.57 SPAD units for Experiments 1 and 2, respectively (Table 1). At 45 DAP no significant difference between the treatments and their controls was observed from Experiment 1, while in Experiment 2, pH and CFS treatments influenced the LG (Table 1). At 45 DAP 71EL2006H∼1% CFS treatment from *L. helveticus* EL2006H cultured at pH 7 had a significantly higher LG (46.63 SPAD units) than both positive controls (Table 1). All the treatment and pH levels exhibited reduced greenness index at 45 DAP.

LA contributes greatly to the physiological performance of plants, relating to photosynthesis, evapotranspiration, and plant growth. There was neither an effect of pH nor CFS treatment in Experiment 1, but in Experiment 2 the interactive effect of potato growth pH and CFS was the contributor to the variation (Table 2).

Photosynthetic rates measurement at 30 and 45 DAP, respectively, varied significantly between pH levels in Experiment 1 in the first 30 DAP. At 45 DAP the effect of CFS treatment was prominent in that 52EB2004S~0.4% CFS obtained from *B. subtilis* EB2004S cultured at pH 5 caused a greater photosynthetic rate (12.33 μmol m^−2^ S^−1^) than the controls and other CFS treatments in Experiment 1 (Table 1). The effect of pH at 45 DAP for Experiment 1 was not significant, as it had been at 30 DAP (Table 1). There were no significant effects of either pH or CFS treatments in Experiment 2 (Table 1).

A two-way ANOVA at 30 DAP for Experiment 1 revealed significantly higher stomatal conductance (0.07 mol m^−2^ S^−1^) at pH 5 compared to that recorded at pHs 7 and 8 (Table 1). However, stomatal conductance was not affected by CFS treatments. Fifteen days later, within the same experiment and in addition to the pH effect, a CFS treatment effect was significant at *p* < 0.001 in that 52EB2004S~0.4% from *B. subtilis* EB2004S cultured at pH 5 resulted in higher stomatal conductance (0.10 mol m^−2^ S^−1^) than positive and negative controls (Table 1). The same CFS resulted in higher values of stomatal conductance than a treatment with *L. helveticus* EL2006H (Table 1). In Experiment 2, CFSs from neither *B. subtilis* EB2004S nor *L. helveticus* EL2006H influenced stomatal conductance; only pH had an effect (Table 1).

Stem diameter and plant height are important growth indices of potato. The current study measured differences in stem diameter as influenced only by pH in Experiment 2 at both 30 and 45 DAP, while no effect of the treatment combination was detected in Experiment 1 (Table 1). Contrasting results were revealed in that at 30 DAP potato grown at pH 8 showed greater stem diameter, while at 45 DAP those from pH 5 had gained more stem diameter. Data analysis on potato plant height in this study indicated no significant effect of CFS treatment in either experiment at 30 or 45 DAP (Table 1).

The comparison between CFS treatments and their controls indicated significant differences at *p* < 0.05, for both experiments. In Experiment 2, SFW was significantly affected by pH, but not CFS. The effect of pH 5 resulted in greater SFW (120.76 g plant^−1^) compared to pH 8 (102.05 g plant^−1^) (Table 2). After drying the shoots, their weights were recorded and analyzed. In Experiment 1, CFS treatments and pH did not affect shoot dry matter (SDM) (Table 2).

The highest root fresh weight (RFW) (56.25 g plant^−1^) resulted from treatment with 52EB2005S~0.4% CFS of *B. subtilis* EB2004S cultured at pH 5, numerically greater than the positive and negative controls, but statistically not different from treatment with 52EL2006H~0.4% CFS of *L. helveticus* EL2006H also cultured at pH 5, which resulted in 55.12 g plant^−1^ (Table 2). The potato growth pH affected (increased) RFW (54.57 g plant^−1^) at pH 7 for Experiment 1 with respect to pHs 5 and 8 (Table 2). Analysis of the data of root dry matter (RDM) showed no statistical differences among the treatment factors in Experiment 1, indicating no possible influence on the variable (Table 2). The whole plant dry matter (WPDM) was affected mainly by pH for Experiment 1, where higher pH 8 recorded the least weight (19.12 g plant^−1^) (Table 2).

### 2.2. The Interactive Effect of pH and CFS Treatments on Potato Growth and Physiological Variables

For Experiment 1, pH and CFS treatment interaction was significant for plant height and leaf greenness/chlorophyll at 30 DAP, SFW, and WPFW (Table 1 and Table 2 and Figure 1A–D). The interaction of 52EB2005S~0.4% CFS with the potato grown at pHs 5 and 8 resulted in a significant (*p* < 0.0001) plant height compared to the positive control receiving the same treatment combination (Figure 1A). Leaf greenness, on the other hand, was significantly influenced by the interaction of potato grown at pH 8 with a 52EL2006H∼0.4% CFS treatment with higher greenness over 51EL2006H∼1% CFS treatment which had a higher concentration but less effect (Figure 1B). Interaction effect also contributed to the variations in SFW and WPFW in Experiment 1 (Figure 1C,D). Furthermore, the interactive effect of pH and CFS treatment contributed significantly to the SFW differences of Experiment 1 (Figure 1C). In Experiment 1, pH, CFS treatments, and their interaction affected WPFW (Table 1, Figure 1D).

In Experiment 2, CFS treatments 72EB2005S~0.4%, 83EB2005S~0.2%, and 71EL2006H~1% were evaluated for their effect on potato growth under greenhouse conditions. The pH and CFS treatment interactions were significant in plant height and leaf greenness/chlorophyll at 30 DAP, leaf area, shoot dry matter, root fresh and dry weight, and whole plant fresh and dry weight (Table 1 and Table 2, Figure 2A–H). Increased plant height was observed because of interaction of potato grown at pH 5 and the 72EB2005S~0.4% CFS treatment compared to all controls. On the other hand, there was a numerically higher plant height influenced by 71EL2006H~1% CFS treatment over all controls. Surprisingly, the interaction of growing potato at pH 7 and 71MRSCONT∼1% resulted in the tallest plants at pH 7 (Figure 2A). Relative leaf greenness had less variation because of pH and CFS treatment interaction as indicated in Figure 2B. Potato shoot dry matter (SDM) level was substantially high at pH 5 for 72EB2005S~0.4% CFS treatment (Figure 2C) as well as the leaf area for the same treatment combination (Figure 2D). There were significant interaction effects on potato RFW in Experiment 2 which involved CFS obtained from *B. subtilis* cultured at pHs 7 and 8 and *L. helveticus* cultured at pH 7 (Figure 2E). RFW was numerically high over the positive control for a combination of pH 8 and 71EL2006H~1% CFS treatment and significantly above the negative control (Figure 2E). Similarly, in Experiment 2 CFS treatments had significant effects on RFW in that 72EL2006H~0.4% resulted in greater (49.29 g plant^−1^) levels than all other treatments, although not statistically different from the positive control (Table 2). Specifically, 83EB2005S~0.2% CFS treatment interacted with potato grown at pH 7, resulting in significantly higher RFW over its positive control, but numerically similar to other treatments and controls. A higher RDM over the positive control was a result of 83EB2005S~0.2% CFS treatment and potato grown at pH 7 (Figure 2F). Consistently, 72EB2005S~0.4% CFS treatment was associated with accumulated higher WPFW and WPDM over the positive control (Figure 2G,H). Furthermore, visual observation of the roots showed the presence of tubers being already formed for the plants at pHs 7 and 8 treated with 52EB2005S~0.4% CFS of *B. subtilis* EB2004S cultured at pH 5 (Figure 3A), while 52EL2006H~1% CFS and 52EL2006H~0.4% CFS did not influence early tuberization (Figure 3B,C).

## 3. Discussion

Ways to avoid the constraints on PGPM inoculation could be the use of CFS in replacement of microbial cells in promoting plant growth and stress tolerance [24,40]. Previous studies have elucidated the importance of pH on the growth and development of potato [41,42]. The effect of PGPMs on enhancing potato growth has been an area of study in recent years considering the use of sustainable means to increase productivity of plants. While many questions remain unanswered regarding the best deployment of PGPMs and PGPM products to ensure effective output, more questions arise even from the unanswered questions. Even though the potential of microbes has not been fully explored with respect to potato growth promotion, the use of live microbes has faced difficulties in attaining their full potential for plant growth stimulation, as reported under both laboratory and controlled environment conditions [39]. The use of CFS from PGPMs, a relatively new approach, has gained interest. A specific focus of this study was to assess the potential of CFS from *B. subtilis* EB2004S and *L. helveticus* EL2006H produced at pHs 5, 7, and 8 applied on potato plants subjected to similar levels of pH.

### 3.1. Effect of CFS Treatment and pH on Leaf Greenness (LG), Leaf Area (LA), Photosynthetic Rate, and Stomatal Conductance

A non-destructive measure of chlorophyll, such as through use of the SPAD meter, helps in monitoring plant performance over time with analysis of a wide range of other plant growth and development variables. Previous studies have shown that habitat pH does not influence leaf greenness for *Spiraea alba* and *Spiraea tomentosa* [43,44]; on the contrary, the current study on potato has indicated that lower pH increases leaf greenness significantly compared to higher pH. It is known that SPAD values are related to nitrogen concentration in plant tissues [45], which could be related to increased solubility of nutrients in the Hoagland solution at lower rather than higher pHs. In the current study SPAD was used to determine the relative greenness of leaves at 30 and 45 DAP as an assessment of response to microbial CFS treatment. The general observation is that during the first 30 DAP the interactive effect of pH and CFS treatment combinations was prominent, but 15 days later at 45 DAP, there was no significant effect of these treatments in Experiment 1, while in Experiment 2, what was prominent was the effect of the main factors of pH and CFS. Previously, leaf greenness in potato has shown direct influences on tuber yield [46,47], suggesting that CFS influence on chlorophyll has potential to influence yield as well, as seemed to be the case in the current work. The reason for the disappearance of the effect could be related to plants being at their vegetative stage initially and having commenced flowering when later measurements occurred. The decrease in the SPAD index values recorded at 45 DAP might be associated with the growth stage of potato (flowering/tuber initiation), which requires more nutrients. Studies in other crops reported increased chlorophyll content as the result of CFS application, such as *Enterococcus faecium* with *Cucumis melo* [48] and *Fusarium tricinctum* RSF-4L with *Oryza sativa* [49]. This report, to the best of our knowledge, provides the first data on potato with respect to the effect of CFS on LG; it is known that at high pH there is usually decreased availability of nutrients from the soil colloids as well as from watering solutions. Specifically, Ca, Fe, Mg, Mn, and P become more limited, which reduces chlorophyll content in plants [50]; this study shows that at pH 8 the leaf greenness SPAD readings were significantly lower than the pH 5 and 7 readings. Furthermore, the apoplast has an average pH of around 5.5, comparatively lower than that of the cytoplasm, which is around 7 [51,52], which creates a gradient allowing flow of nutrients. In cases of higher pH the root apoplast pH increases, causing disruption of the gradient, hence impairing nutrient translocation [53].

With regard to LG, the LA is an important plant variable that is responsible for how much light is intercepted for photosynthesis and how much water can escape through evapotranspiration and ultimately contributes meaningfully to plant growth [54]. Like many plant ecology and physiology investigations, this study assessed the effects of CFS and pH on LA as it has been a good estimator of plant nutrition and ultimately assists in understanding plant growth and yield potential [55,56]. This study indicated that 72EB2004S~0.4% CFS treatment significantly increased LA in potato grown at pH 5 (Figure 2D). Even though the underlying mechanisms of pH and CFS treatment interaction for the current findings are not directly clear, it could suggest that similar mechanisms used by live PGPMs might be responsible for increased LA [52,57]. The use of CFS as a biostimulant has been increasing owing to the resulting enhancement of plant growth and development [36,40].

Photosynthetic rate is extremely important in growth and development of plants. This variable is influenced by many factors including the time of the day, chlorophyll content, LA, LG, number of optimal conditions the plant is experiencing at a particular time, and stage of growth of the plant. The pH of the medium in which a plant is growing influences nutrient availability, potentially leading to insufficient, optimum, and excessive availabilities [58], which translates to proper and improper plant growth and performance if variations are significant. However, the response of plants to pH variation differs among plant species as does their photosynthetic rate [58]. In the current study, at 45 DAP the effect of CFS treatment was prominent, in that 52EB2004S~0.4% CFS treatment resulted in a higher photosynthetic rate (12.33 μmol m^−2^ S^−1^) than the controls and other *L. helveticus* EL2006H CFS treatments. While this is the first time CFS from *B. subtilis* EB2004S cultured at suboptimal pH 5 has been applied on potato to observe its effect on photosynthesis, a comparatively similar approach has been reported on the effect of fungal CFS of *Penicillium citrinum* KACC43900, having resulted in increased photosynthesis following a foliar spray on *Carex kobomugi* [59].

A close relationship between the photosynthetic rate and stomatal conductance took the same trajectory with respect to CFS treatment on potato. Significantly higher stomatal conductance (0.07 mol m^−2^ S^−1^) at pH 5 compared to that recorded at pHs 7 and 8 for Experiment 1 (Table 1) was the first indicator that hydrogen ion (H^+^) concentration has an impact on this variable. With the relationship between photosynthetic rate and stomatal conductance having been broadly studied, including for potato [60], it is not surprising to find a similar effect of 52EB2004S~0.4% CFS treatment resulting in higher stomatal conductance levels (0.10 mol m^−2^ S^−1^) than positive and negative controls.

### 3.2. Effect of CFS Treatment and pH on Potato Plants’ Stem Diameter and Height

Plant stem diameter and height are the best indicators of plant vigor and growth, including in potato. Stem diameter in the current study at 30 DAP was only influenced by CFS treatment obtained from pHs 7 or 8 only in Experiment 2 (Table 1). Potato stem diameter is known to have positive correlation with number of leaves which together contribute to number of tubers produced [61]. In this study, knowing that factors such as tuber size influence plant height [62], uniformity of tubers before sowing was considered to ensure uniformity at the beginning of treatment application. The use of CFSs from *Bacillus* sp. CaSUT007 and *Bacillus subtilis* EA-CB0575 increased shoot length for *Manihot esculanta* and *Musa* spp. [63,64]. Interestingly, the application of 52EB2004S~0.4% CFS treatment on potato grown at pHs 5 and 8 resulted in significantly (*p =* 0.0009) higher plant height than its positive control, but similar heights to the negative control. Moreover, the interaction of potato grown at pH 5 with a 72EB2004S~0.4% CFS treatment caused significantly higher plant height than the negative control, and a height numerically above the positive control. This is, most likely, the first report of CFS treatments of the same microbe cultured at different pH conditions contributing to enhanced potato plant height. The main effector compounds contributing to such plant height enhancement have been associated with indole acetic acid (IAA) and extracellular proteins [63,64] although specific microbe to plant signals are possible [28]. The protein profiling of the strains’ CFSs used in this study revealed the presence of proteins with diverse functions, including some with unknown functions [65] potentially supporting the observed effects. Specifically, *B. subtilis* EB2004S CFS obtained from pH 5 secreted chain B selenosubtilisin, the only upregulated protein [65], which might have been crucial in protecting potato root cells from pH damage, resulting in enhanced plant growth. This is further emphasized by the increased biological activities related to cellular detoxification, cell division, and response to stress only expressed with CFS of *B. subtilis* EB2004S obtained at pH 5, which indicated selenosubtilisin protein was involved. Furthermore, the fact that selenosubtilisin is involved in amide hydrolysis [66] might have increased the presence of ammonia in the CFS and hence increased available nitrogen for plant uptake which resulted in enhanced plant growth.

### 3.3. Effect of CFS Treatment and pH on Potato Shoot Fresh Weight (SFW), Shoot Dry Matter (SDM), Root Fresh Weight (RFW), and Root Dry Matter (RDM)

The increase in biomass for plants is a part of increased plant growth. Application of microbial CFSs has been reported to improve plant biomass accumulation and stress tolerance [67]. Studies using CFS have identified notable phytohormones potentially responsible for plant growth promotion, mostly IAA, while extracellular proteins, peptides, and siderophores are also known to play important roles. A study by Buensateai, Sompong, Thamnu, Athinuwat, Brauman, and Plassard [63] showed that CFS from *Bacillus* sp. CaSUT007 had significant effects in increasing shoot and root length and ultimate whole plant biomass of cassava. Moreover, application of CFS from *B. subtilis* EA-CB0575 on banana plants resulted in increased plant dry weight [64]. In this study, the highest SFW was recorded in potato grown at pH 5 treated with 52EL2006H~0.4% CFS, followed by those grown with a combination effect of pH 8 and a 52EB2004S~0.4% CFS treatment (Figure 1D). This is also reflected in previous studies indicating that CFS can increase plant fresh and dry weight, however, variation in pH conditions during culture of the microbes is a contributing factor.

The effects of pH in the current study are in accordance with a previous study which showed potato at pH 5 had more biomass than those grown at pHs lower than 5 and higher than 7 [68]. The findings of this study are consistent with previous work using the same CFSs on tomato and corn seedling growth variables [40] which involved the same strains and same concentrations of the CFSs. Moreover, the visual observation of the roots showed potato treated with a 52EB2004S~0.4% CFS treatment grown at pHs 7 and 8 had tubers, indicating that the CFS might have influenced early tuberization. The effect of rhizobacteria on early tuberization has been reported to be through enhanced lipoxygenase activities [69], but the mechanisms of CFS activity are yet to be identified in this regard. However, the CFS of *B. subtilis* EB2004S cultured at pH 5 as reported previously in Msimbira et al. [65] had increased ABC-F family ATP-binding-cassette proteins which are usually responsible for solute transport around membranes and stress tolerance, leading to enhanced growth and to early tuberization. However, selective activity on early tuberization of as a result of 52EB2005S~0.4% CFS treatment for potato grown at pHs 7 and 8 and not at pH 5 requires further confirmation.

Roots are important plant organs and, apart from being responsible for water and nutrient acquisition, they also function by anchoring the plant in soil or growth medium [70]. In potato, the functions are further extended to tuber formation from special belowground structures (stolons) [71,72] as opposed to grains. Root fresh weight was influenced by 52EB2005S~0.4% CFS treatment which resulted in the relatively high weight of 56.25 g plant^−1^ (Table 1). The interactive effect of potato grown at pH 7 in combination with application of 83EB2004S~0.4% CFS treatment resulted in increased RFW and RDM (Figure 2E,F). That CFS caused increases in root fresh and dry weight was previously reported for wheat through pre-treatment of seeds with *Streptomyces atroolivaceus* CFS [73]. The increase in RFW and RDM could be a result of the bioactivity of extracellular proteins found in the CFS, as previously identified in a secretome analysis of these strains [65]; they could, however, also be the result of interaction with pH among other unknown compounds, factors requiring further investigation. Extracellular proteins, being related to root biomass increase, are also reported in *Glycine max* after treatment with CFS from *Bacillus amyloliquefaciens* KPS46 [74]. Protein profiles of CFS for *B. subtilis* EB2004S cultured at pH 5 expressed increased clusters of ABC-F family ATP-binding-cassette proteins, a family of soluble proteins within the ABC superfamily [65]. These proteins are involved in protein synthesis when cells encounter multiple stresses. In plants the ABC proteins are responsible for transmembrane molecule transport of phytohormones, organic acids, and metal ions, contributing to enhanced plant growth and development [75]. The current study, therefore, provides primary information that culturing the same microbe at different pH levels could be beneficial depending on the final use, such as the expression of ABC proteins which were only expressed under pH 5.

### 3.4. Effect of CFS Treatment and pH on Potato Plant WPFW and WPDM

Plant biomass accumulation is a direct measure of growth. The partitioning of biomass to harvestable parts determines the harvest index of the crop (tubers in the case of potato). Since potato tubers were not fully developed, calculating the harvest index at 45 DAP was not possible, hence the whole plant biomass was considered in the assessment of the effect on overall plant biomass accumulation of CFS treatments. The findings obtained in the current study show significant increases in WPFW and WPDM, confirming that there were effects of CFS application and/or partly interactive effects of pH and CFS treatments, and therefore justify rejecting the null hypothesis. Potato plants growing at pH 5 treated with a 52EL2006H~0.4% CFS treatment recorded a mean WPFW above the controls (Figure 1D). Similarly, an interactive effect of pH 8 and a 52EB2004S~0.4% CFS treatment increased WPFW significantly over the positive control (Figure 1D). This was consistent with the previous study on seedling growth of *Solanum lycopersicum* and *Zea mays*, showing they were influenced by application of the same CFS of the same strains [40]. A study on *Cucumis sativus* using CFS from *Cladosporium* sp. showed increases in both plant fresh and dry weights due to the effects of gibberellins found in the CFS [76]. Potato plants that received a treatment combination involving 83EB2004S~0.2% CFS treatment had greater WPDM than the positive control. The effect of CFS on plant dry weight has been reported in other crops such as *Cucumis sativus* [76] and *Coreandrum sativum* [77].

Though the study has not clearly identified specific mechanisms contributing to such effects, it has provided clear evidence that is consistent with previous studies on *Solanum lycopersicum* and *Zea mays* conducted using the same strains’ CFSs [40] and featuring variability of secretome proteins of *L. helveticus* and *B. subtilis* [65]. CFSs, especially proteins of unknown functions within the CFS, might also be the contributing factors. The presence of peptides from CFS responsible for plant growth promotion may also be considered as contributing factors as was reported from *Bacillus thuringiensis* NEB17 commercialized as Thuricin 17 [78,79]. The findings of this study prove that using a pre-selected optimum benchmark pH is less effective than varying pH levels when it comes to both culturing microbes and applying them to plant growth media; furthermore, replacing plant growth-promoting microbes with cell-free supernatants extracted from PGPMs at specific pH levels may be more beneficial to plant growth than the PGPM inoculation itself.

## 4. Materials and Methods

### 4.1. Preparation of CFS Treatments

The microbial strains used in this study were obtained from EVL Inc. (http://www.evlbiotec.com, accessed on 10 February 2023), as a subset of a microbial biostimulant consortium in 2018. The two bacteria, *B. subtilis* EB2004S and *L. helveticus* EL2006H, were selected based on previous reported effect of their CFSs on enhancement of seed germination and seedling growth for corn and tomato [40]. The strains were revived from working stock cultures stored at −20 °C; a loopful was added into 50 mL fresh sterile M13 (*B. subtilis* EB2004S) or MRS (*L. helveticus* EL2006H) medium broth, contained in 250 mL flasks. The cultures were then incubated for 48 h at 30 and 37 °C for *B. subtilis* EB2004S and *L. helveticus* EL2006H, respectively. An additional incubation condition including the use of an orbital shaker at 120 rpm was used for *B. subtilis* EB2004S, while *L. helveticus* EL2006H was incubated without agitation.

To obtain CFSs, strain cultures after 48 h of incubation were transferred to sterile Falcon tubes for cell separation by centrifugation at 12,000× *g*. The supernatant was filtered using vacuum filters (0.22 µm pore size) (Corning^®^ Vacuum Filter systems, Fisher Scientific, Tewksbury, MA, USA) to ensure no cells or debris remained. The obtained CFSs from pHs 5, 7, and 8 for *B. subtilis* EB2004S and pHs 5 and 7 for *L. helveticus* EL2006H, including sterile medium (positive control) at respective pHs and a negative control, were used for treatments as indicated in Table 3. For final treatment formulations, a ratio of CFS to distilled water (volume of CFS/volume of distilled water) was used to obtain 1, 0.4, and 0.2% (Table 3).

### 4.2. Experimental Design and Layout

Greenhouse experiments were carried out at the Macdonald Campus of McGill University between winter 2021 and winter 2022. Greenhouse pot experiments involved potato (*Solanum tuberosum* L., variety Goldrush) as the host bioassay plant. This variety was selected because of its popularity in the province of Quebec, and due to the positive results when treated with EVL biostimulant products, including the two strains of the consortium strains used in this study. CFSs from *B. subtilis* EB2004S and *L. helveticus* EL2006H were evaluated for their effects on potato growth under greenhouse conditions. Medium-sized (3–4 cm) potato seed tubers were grown in 8.82 L pots. A total of 63 pots were used, arranged following a completely randomized design (CRD), with the CFS treatments of the experiments (Table 1) assigned randomly to 21 pots corresponding to three sets of pH levels (5, 7, and 8). Each complete set of the treatments had three replications. Two factors (pH levels (5, 7, and 8) and CFSs from *B. subtilis* EB2004S and *L. helveticus* EL2006H obtained from pH 5) were used in the first experiment (Experiment 1), while the second experiment (Experiment 2) constituted CFS obtained from pHs 7 and 8 (Table 1). Limited greenhouse space led to treatment subdivision based on the pH level of origin of the CFS. The choice to make the division based on pH of the CFS production rather than strain was because of the desire to compare how CFS of *B. subtilis* EB2004S and *L. helveticus* EL2006H originating from the same pH compares with respect to potato growth promotion. The organization of the resulting two experiments is indicated in Table 1. The entirety of each experiment was repeated twice.

### 4.3. Treatment Application and Experiment Management

There were two sequential CFS treatments, one at planting and the second at 30 days after planting (DAP), along with one zero CFS control treatment as a negative control and sterile broth medium as a positive control (Table 1). The CFS treatments were applied twice at the beginning and at 30 DAP to increase the chances of capturing their effect at early vegetative and flowering growth stages. Three pH levels (5, 7, and 8), which constituted the second factor through water and nutrient solutions, were maintained throughout the experiment with three replications for each pH by CFS treatment combination. The first CFS treatment application was applied by soaking potato seed overnight in treatment solution before planting in pots. A total of 9 selected potato seed tubers with 2 or more eye buds were soaked for each treatment then sown in 8.82 L pots containing 500 g of perlite. Perlite was used in this experiment because it has a neutral pH which corresponds to the reference pH of the treatments. Furthermore, perlite has high moisture retention, aeration, and no nutrients which allowed equal supply of nutrient from the nutrient solution. After sowing, the pots were placed on top of benches following a pre-randomized order and watered with 100 mL of decanted fluid from the overnight soaking, followed by 1 L of water at adjusted pHs of 5, 7, and 8 using HCl or KOH. The greenhouse conditions were set at 16/8 h light/dark and 23 ± 2/18 ± 2 °C day/night temperature with 68 ± 2% relative humidity (RH). Pots were watered to saturation on the planting date followed by watering once to twice a week with respect to moisture balance. All the pots received equal volumes of treatment nutrient solution and water at each watering. Half-strength Hoagland solution [80] was prepared with the composition as previously described in Msimbira et al. [38] and its pH adjusted to 5, 7, and 8 and used for watering plants alternating with water in the first 30 DAP. A second application of the treatment was applied on the 30th DAP by mixing CFS in half-strength Hoagland solution at respective concentrations for each treatment, then dispensing 150 mL per pot. Plants were then irrigated with full-strength Hoagland solution and water interchangeably, as required.

### 4.4. Data Collection

Potato plants were managed from sowing to 30 DAP, at which time the first non-destructive data collection was made. Potato physiological variable indices were measured using a portable photosynthesis system (Li-Cor 6400, Lincoln, NE, USA) with photosynthetic photon flux density set at 800 μmolm^−2^ S^−1^, CO_2_ concentration of 400 ppm or μmol mol^−1^, flow rate of 500 μmol S^−1^ in the chamber was used to measure photosynthetic rates and stomatal conductance of fully expanded leaves of potato plants at 30 and 45 DAP. The chamber temperature and RH were set at 25 °C and 44%, respectively. The relative leaf greenness was measured using a SPAD meter (chlorophyll meter) (SPAD-502, Konica Minolta Sensing Inc., Tokyo, Japan) at 600 nm. Potato growth indices (plant height measured using a ruler at 0.1 cm accuracy and stem diameter measured using vernier calipers at 0.001 mm accuracy) were also recorded at both 30 and 45 DAP. The second treatments were applied on the day of first data collection to each pot. Plants were then left to grow for another 15 days and then harvested at 45 DAP. Careful cutting of potato shoots at the soil (perlite) line was carried out using a pruner, separating the potato plants into shoots and roots. The leaves from the shoots were detached and scanned for their total area using a leaf area meter (LI-3100 C, Lincoln, NE, USA), after which fresh shoot (stems and leaves) weight was also recorded. Roots were then carefully removed from perlite growth medium by gentle shaking and washed to remove the loosely adhered perlite particles. The roots were aerated on blotter paper to remove the excessive water from washing, then fresh weight was recorded. Roots and shoots, in separate envelopes, were then dried to a constant weight in an oven set at 60 °C, to determine dry matter production.

### 4.5. Data Analysis

The data collected from the greenhouse experiments were tested for normality to fulfil statistical assumptions before being subjected to analysis of variance (ANOVA) using PROC GLM of SAS 9.4 for Windows (SAS Institute, Inc., Cary, NC, USA). Two-way ANOVA was performed to determine main effects of CFS treatments and pH (5, 7, and 8) and interaction effects for the two factors (pH and CFS treatments) on response variables, leaf greenness, leaf area, plant height, photosynthesis, stomatal conductance, shoot fresh and dry weight, root fresh and dry weight, and whole plant fresh and dry weight. Treatment means specific differences were tested using Fisher least significant difference (LSD) at α = 0.05. Graphs of mean values and results tables were generated using Microsoft Excel (Microsoft 365, Version 2301).

## 5. Conclusions

The findings of this study are unique in that it probed the effect of CFS produced under differing pH conditions (some of which are relevant to commercial bioreactor production), compared to the standard practices of producing the microbes under optimum growth conditions and applying them to mitigate stresses in plants. While it is not possible to state the mechanisms involved in the effects observed, it is interesting to note that pH and pH-dependent CFS had positive effects on most of the studied potato variables, including chlorophyll, photosynthesis, and stem diameter at 30 DAP. Although the results of non-destructive data collected at 30 DAP did not remain consistent to the final harvest day (45 DAP), this study has been very insightful to learn the effect of CFS at the transition of potato growth stages. Visual observation of the roots showed potato treated with a 52EB2004S~0.4% CFS treatment grown at pHs 7 and 8 had tubers, an indication that CFS influenced early tuberization. In conclusion, there is a need for further research to clarify the effect of CFSs, however, the findings reported here are highly encouraging. Determination of both microbial CFS and potato plant cellular metabolic profiles could further fill in the gaps in scientific understanding.

## Figures and Tables

**Figure 1 ijms-24-06620-f001:**
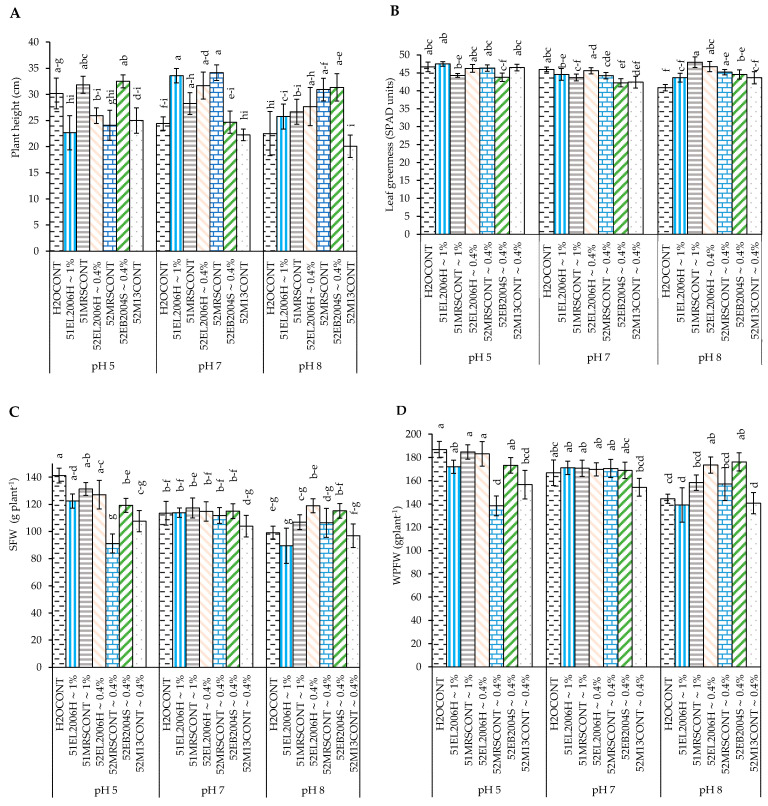
The interaction effect of pH and CFS treatments obtained from the cultured *L. helveticus* and *B. subtilis* (Experiment 1) at pH 5 on (**A**) potato plant height; (**B**) leaf greenness measured in SPAD units; (**C**) shoot fresh weight (SFW*); and (**D**) whole plant dry matter (WPDM*) under greenhouse conditions. Values are expressed in their respective units and means, dissimilar letters indicate significant (*p* = 0.05, LSD) differences among treatments. Error bars represent standard error of means (SEM).

**Figure 2 ijms-24-06620-f002:**
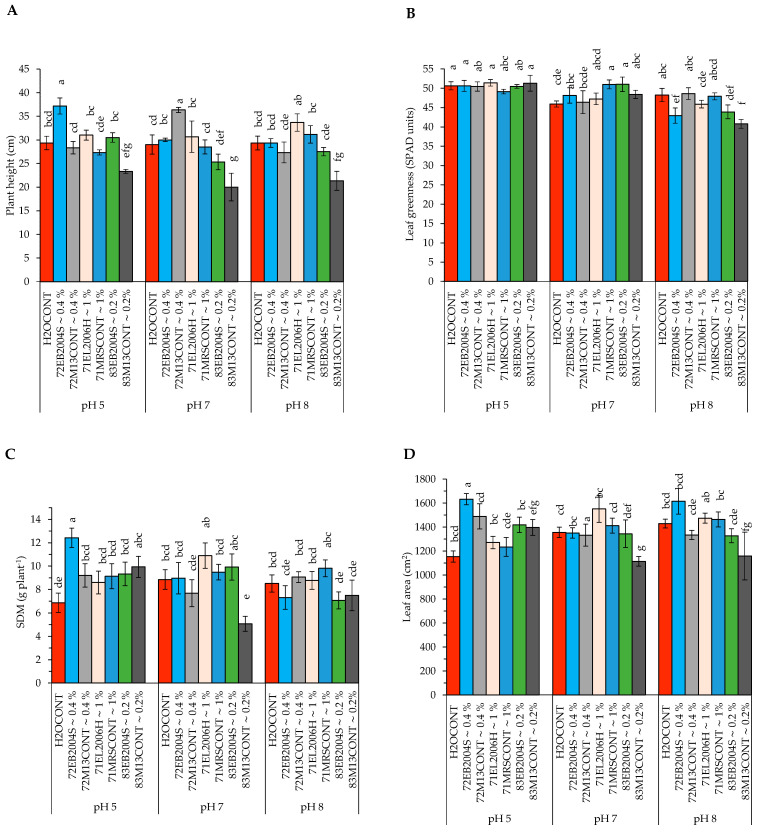
The interaction effect of pH and CFS treatments obtained from the cultured *L. helveticus* and *B. subtilis* (Experiment 2) at pHs 7 and 8 on potato; (**A**) plant height; (**B**) leaf greenness measured in SPAD units; (**C**) leaf area; (**D**) shoot dry matter (SDM*); (**E**) root fresh weight (RFW*); (**F**) root dry matter (RDM*); (**G**) whole plant fresh weight (WPFW); and (**H**) whole plant dry matter (WPDM*) under greenhouse conditions. Values are expressed in their respective units and means, dissimilar letters indicate significant (*p* = 0.05, LSD) difference among treatments. Error bars represent standard error of means (SEM).

**Figure 3 ijms-24-06620-f003:**
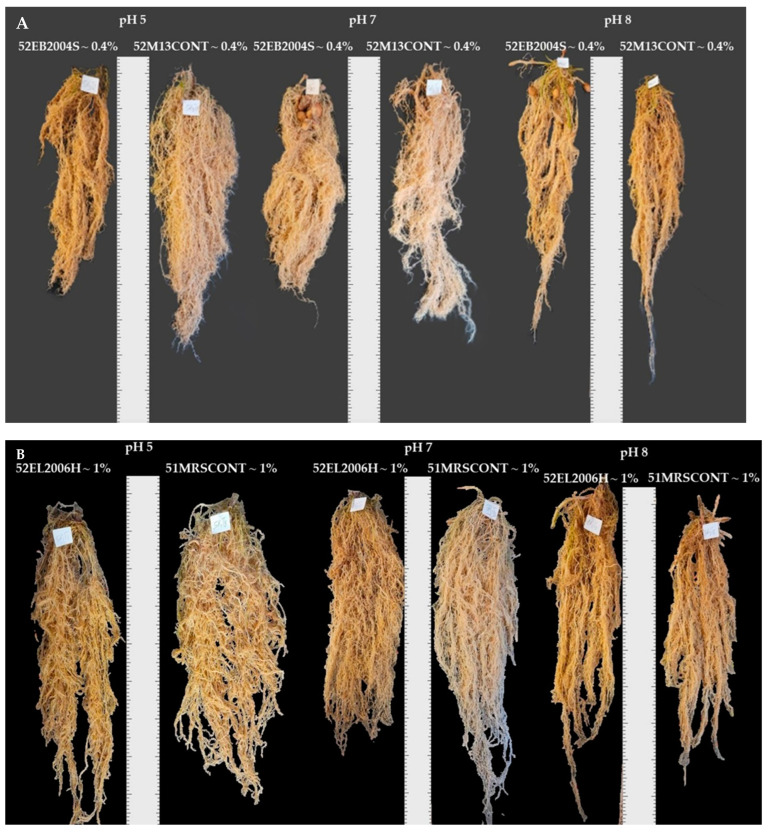
Visual variations of potato roots from plants grown at pHs 5, 7, and 8 in response to CFS from *B. subtilis* EB2004S cultured at pH 5 (**A**) and *L. helveticus* EL2006H (**B**,**C**).

**Table 1 ijms-24-06620-t001:** A two-way ANOVA on the response of potato to CFS and pH treatments on selected growth (stem diameter and plant height) and physiological (chlorophyll, photosynthesis, and stomatal conductance) variables under greenhouse conditions. Experiment 1 represents CFS treatments obtained at pH 5; Experiment 2 represents CFS treatments obtained at pHs 7 and 8.

Experiment	Factors	Stem Diameter		Plant Height		Chlorophyll		Photosynthesis		Stomatal Conductance	
**1**	**CFS Treatments**	**(mm)**		**(cm)**		**SPAD Units**		**(μmol m^−2^ sec^−1^)**		**molm^−2^s^−1^**	
		30 DAP	45 DAP	30 DAP	45 DAP	30 DAP	45 DAP	30 DAP	45 DAP	30 DAP	45 DAP
	H_2_OCONT	7.15 ± 0.30	8.39 ± 0.368	25.7 ± 1.6 a	47.9 ± 2.0	44.49 ± 0.8	43.32 ± 0.7	8.08 ± 0.2 b	10.66 ± 0.61 bc	0.06 ± 0.004	0.06 ± 0.003 bc
	51EL2006H∼1%	7.30 ± 0.42	8.37 ± 0.456	27.0 ± 1.5 a	51.5 ± 1.9	45.23 ± 0.7	44.07 ± 0.76	8.41 ± 0.4 ab	11.23 ± 0.63 ab	0.06 ± 0.006	0.07 ± 0.006 b
	51MRSCONT∼	7.67 ± 0.24	8.33 ± 0.361	28.9 ± 1.2 a	51.4 ± 2.2	45.31 ± 0.7	44.45 ± 0.47	7.68 ± 0.3 b	9.88 ± 0.53 bc	0.05 ± 0.004	0.04 ± 0.003 c
	52EL2006H∼0.4%	7.64 ± 0.25	8.63 ± 0.364	28.4 ± 1.2 a	49.7 ± 2.1	46.18 ± 0.6	45.93 ± 0.35	8.49 ± 0.4 ab	10.78 ± 0.56 abc	0.06 ± 0.009	0.06 ± 0.004 bc
	52MRSCONT∼Positive control	7.39 ± 0.30	8.34 ± 0.365	29.7 ± 1.5 a	52.1 ± 2.6	45.25 ± 0.5	43.76 ± 0.62	7.97 ± 0.3 b	9.42 ± 0.33 c	0.06 ± 0.007	0.05 ± 0.003 c
	52EB2004S∼0.4%	7.45 ± 0.29	8.55 ± 0.188	29.5 ± 1.4 a	50.6 ± 1.9	43.51 ± 0.6	44.42 ± 0.66	9.27 ± 0.4 a	12.33 ± 0.7 a	0.06 ± 0.006	0.1 ± 0.011 a
	52M13CONT∼Positive control	8.02 ± 0.36	8.60 ± 0.344	22.4 ± 1.1 b	45.2 ± 2.4	44.18 ± 0.8	44.35 ± 0.55	8.12 ± 0.2 b	9.69 ± 0.45 bc	0.06 ± 0.004	0.05 ± 0.003 bc
	pH										
	5	7.81 ± 0.20	8.74 ± 0.158	28.4 ± 0.9	50.2 ± 1.2	45.89 ± 0.3 a	44.44 ± 0.34	9.03 ± 0.2 a	10.57 ± 0.37	0.07 ± 0.004 a	0.07 ± 0.005 a
	7	7.37 ± 0.18	8.43 ± 0.246	27.5 ± 0.9	50.1 ± 1.5	44.09 ± 0.5 b	44.05 ± 0.45	8.08 ± 0.2 b	10.32 ± 0.38	0.05 ± 0.003 b	0.06 ± 0.003 ab
	8	7.37 ± 0.22	8.20 ± 0.266	26.4 ± 1.0	49.1 ± 1.5	44.65 ± 0.4 b	44.49 ± 0.41	7.75 ± 0.2 b	10.83 ± 0.4	0.05 ± 0.003 b	0.05 ± 0.003 b
	*p*-values										
	**pH**	0.2868	0.2866	0.297	0.8354	**0.0136**	0.6955	**0.0004**	0.6297	**0.0046**	**0.0121**
	**CFS Treatments**	0.6784	0.9931	**0.0033**	0.33	0.1168	0.1006	0.0537	**0.0066**	0.9453	**<0.0001**
	**Interaction**	0.3251	0.942	**0.0009**	0.7959	**0.0047**	0.3081	0.5901	0.6361	0.8756	0.5838
**2**	H_2_OCONT∼Negative control	9.67 ± 0.22	9.3788 ± 0.353	29.22 ± 0.9 bcd	55.7 ± 1.0	48.27 ± 0.82	46.77 ± 0.53 a	12.3 ± 0.60	12.8 ± 0.66	0.09 ± 0.009	0.09 ± 0.01
	72EB2004S∼0.4 %	10.17 ± 0.23	10.13 ± 0.284	32.16 ± 1.05 a	55.8 ± 1.6	47.22 ± 1.25	45.52 ± 0.48 abc	12.2 ± 0.68	13.19 ± 0.66	0.09 ± 0.012	0.09 ± 0.009
	72M13CONT∼Positive control	9.8 ± 0.29	9.8077 ± 0.317	30.66 ± 1.27 abc	54.4 ± 1.5	48.48 ± 1.17	44.59 ± 0.45 c	12.42 ± 0.59	13.91 ± 0.70	0.09 ± 0.010	0.09 ± 0.008
	71EL2006H∼1 %	10.03 ± 0.30	9.7177 ± 0.396	31.77 ± 1.27 ab	56.9 ± 1.9	48.18 ± 0.84	46.63 ± 0.38 a	12.94 ± 0.68	13.73 ± 0.61	0.08 ± 0.008	0.09 ± 0.016
	71MRSCONT∼Positive control	9.82 ± 0.20	9.5727 ± 0.303	29 ± 0.85 cd	56.0 ± 1.4	49.35 ± 0.58	46.22 ± 0.74 ab	11.64 ± 0.63	12.85 ± 0.60	0.08 ± 0.008	0.09 ± 0.01
	83EB2004S∼0.2 %	10.18 ± 0.24	10.052 ± 0.314	27.77 ± 0.83 d	55.2 ± 1.1	48.46 ± 1.14	46.36 ± 0.42 ab	11.99 ± 0.68	14.54 ± 0.62	0.09 ± 0.012	0.12 ± 0.01
	83M13CONT∼Positive control	9.21 ± 0.17	9.26 ± 0.216	21.55 ± 1.16 e	50.7 ± 1.6	46.83 ± 1.34	44.93 ± 0.61 bc	12.08 ± 0.54	14.03 ± 0.70	0.09 ± 0.01	0.11 ± 0.012
	pH										
	5	9.53 ± 0.12 b	10.2 ± 0.198 a	29.57 ± 0.72	54.7 ± 1.0	50.57 ± 0.43 a	46.94 ± 0.33 a	12.45 ± 0.39	12.87 ± 0.47	0.11 ± 0.008 a	0.08 ± 0.006 b
	7	9.79 ± 0.13 ab	9.42 ± 0.184 b	28.54 ± 1.01	55.2 ± 0.9	48.3 ± 0.68 b	45.52 ± 0.33 b	12.42 ± 0.33	13.57 ± 0.41	0.08 ± 0.005 b	0.09 ± 0.006 b
	8	10.2 ± 0.2 a	9.48 ± 0.221 b	28.52 ± 0.8	55.0 ± 1.1	45.47 ± 0.67 c	45.12 ± 0.35 b	11.8 ± 0.49	14.3 ± 0.36	0.08 ± 0.007 b	0.11 ± 0.008 a
	*p*-values										
	**pH**	**0.0095**	**0.0166**	**0.4053**	**0.9415**	**<0.0001**	**0.0006**	**0.4956**	**0.0555**	**0.0162**	**0.0025**
	**CFS Treatments**	**0.063**	**0.4053**	**<0.0001**	**0.1222**	**0.4699**	**0.0162**	**0.9004**	**0.394**	**0.6465**	**0.3426**
	**Interaction**	**0.4793**	**0.9914**	**0.0003**	**0.598**	**0.0043**	**0.8664**	**0.9991**	**0.2057**	**0.9451**	**0.079**

Columns indicate mean values (±SE). Dissimilar letters in a column represent significant variation (*p* = 0.05, LSD) and bold numbers are significant *p*-values.

**Table 2 ijms-24-06620-t002:** A two-way ANOVA on the response of potato to CFS and pH treatments on leaf area, shoot fresh weight (SFW*), shoot dry matter (SDM*), root fresh weight (RFW*), root dry matter (RDM*), whole plant fresh weight (WPFW*), and whole plant dry matter (WPDM*) under greenhouse conditions. Experiment 1 represents CFS treatments obtained at pH 5; Experiment 2 represents CFS treatments obtained at pHs 7 and 8.

Experiment	Factors	Leaf Area	SFW	SDM	RFW	RDM	WPFW	WPDM
	CFS Treatments	(cm^2^)	(g Plant^−1^)	(g Plant^−1^)	(g Plant^−1^)	(g Plant^−1^)	(g Plant^−1^)	(g Plant^−1^)
1	H_2_OCONT∼Negative control	1548.23 ± 114.11	117.89 ± 5.53 a	12.15 ± 0.51	48.17 ± 1.83 b	7.77 ± 1.08	166.07 ± 5.98 abc	19.93 ± 1.12
	51EL2006H∼1%	1657.10 ± 76.03	108.63 ± 5.63 ab	12.53 ± 0.96	52.1 ± 1.93 ab	7.89 ± 0.93	160.74 ± 6.44 bcd	20.42 ± 0.66
	51MRSCONT∼Positive control	1539.28 ± 98.13	118.55 ± 4.01 a	13.00 ± 0.57	52.76 ± 1.28 ab	8.55 ± 1.31	171.31 ± 4.44 ab	21.56 ± 1.09
	52EL2006H∼0.4%	1614.25 ± 86.47	120.32 ± 4.45 a	13.55 ± 0.71	55.12 ± 1.43 a	8.8 ± 1.27	175.45 ± 4.57 a	22.36 ± 1.26
	52MRSCONT∼Positive control	1456.31 ± 110.13	103.13 ± 4.88 b	11.69 ± 0.87	52.27 ± 2.18 ab	8.3 ± 1.21	155.40 ± 6.57 cd	19.99 ± 1.00
	52EB2004S∼0.4%	1612.54 ± 78.82	116.52 ± 2.91 a	12.58 ± 0.55	56.25 ± 2.29 a	8.97 ± 1.25	172.78 ± 4.00 ab	21.56 ± 1.17
	52M13CONT∼Positive control	1380.04 ± 114.75	102.82 ± 4.55 b	12.12 ± 0.77	47.76 ± 2.22 b	7.8 ± 1.04	150.58 ± 5.62 d	19.92 ± 0.74
	pH							
	5	1642.81 ± 60.26	120.03 ± 3.35 a	12.86 ± 0.47	50.67 ± 1.35 b	9.09 ± 0.8	170.70 ± 3.89 a	21.95 ± 0.72 a
	7	1509.37 ± 72.86	112.91 ± 2.43 a	13.00 ± 0.45	54.57 ± 1.04 a	8.38 ± 0.74	167.48 ± 2.79 a	21.39 ± 0.64 a
	8	1479.70 ± 57.16	104.72 ± 3.22 b	11.70 ± 0.47	50.95 ± 1.43 b	7.42 ± 0.68	155.67 ± 4.07 b	19.12 ± 0.58 b
	*p*-values							
	**pH**	0.1844	**0.0007**	0.1033	**0.0469**	0.3325	**0.0041**	**0.0069**
	**CFS Treatment**	0.4787	**0.0062**	0.6183	**0.0117**	0.9855	**0.0034**	0.4452
	**Interaction**	0.9912	**0.0203**	0.4197	0.3315	0.9972	**0.0265**	0.5673
2	H_2_OCONT∼Negative control	1312.98 ± 36.47 bc	108.28 ± 4.86	8.07 ± 0.48	44.85 ± 1.65 b	3.02 ± 0.17 ab	153.15 ± 5.57 a	11.13 ± 0.61 ab
	72EB2004S∼0.4 %	1532.2 ± 49.84 a	117.1 ± 7.66	9.56 ± 0.77	44.84 ± 1.1 b	3.01 ± 0.15 ab	161.96 ± 8.29 a	12.60 ± 0.88 a
	72M13CONT∼Positive control	1384.71 ± 48.74 b	111.27 ± 6.42	8.65 ± 0.52	45.1 ± 1.15 b	3.03 ± 0.14 ab	156.4 ± 7.02 a	11.71 ± 0.63 ab
	71EL2006H∼1 %	1431.94 ± 49.87 ab	120.53 ± 5.61	9.42 ± 0.57	49.29 ± 1.15 a	3.27 ± 0.16 a	169.83 ± 5.53 a	12.72 ± 0.68 a
	71MRSCONT∼Positive control	1369.52 ± 44.06 b	115.98 ± 3.92	9.47 ± 0.45	47.8 ± 1.72 ab	3.33 ± 0.19 a	163.81 ± 4.3 a	12.84 ± 0.56 a
	83EB2004S∼0.2 %	1362.88 ± 46.11 b	111.56 ± 5.50	8.77 ± 0.6	47.18 ± 1.54 ab	3.43 ± 0.14 a	158.76 ± 6.27 a	12.21 ± 0.72 a
	83M13CONT∼Positive control	1222.79 ± 73.1 c	95.17 ± 7.24	7.5 ± 0.72	40.3 ± 1.66 c	2.63 ± 0.16 b	135.51 ± 7.87 b	10.15 ± 0.85 b
	pH							
	5	1370.59 ± 33.57	120.76 ± 3.54 a	9.35 ± 0.40	44.62 ± 0.89	3.19 ± 0.09	165.4 ± 4.02 a	12.57 ± 0.48
	7	1351.12 ± 33.30	111.43 ± 3.91 ab	8.69 ± 0.44	44.97 ± 0.97	2.95 ± 0.10	156.42 ± 4.55 ab	11.68 ± 0.52
	8	1399.87 ± 39.07	102.05 ± 4.14 b	8.29 ± 0.33	47.29 ± 1.13	3.16 ± 0.12	149.37 ± 4.54 b	11.47 ± 0.41
	*p*-values							
	**pH**	0.5307	**0.0021**	0.1084	**0.0614**	0.1991	**0.0199**	0.1551
	**CFS Treatment**	0.0008	0.0505	0.0616	**0.0002**	**0.0083**	**0.0061**	**0.0393**
	**Interaction**	0.00258	0.0946	**0.0008**	**0.0014**	**0.0203**	**0.0277**	**0.0009**

Columns indicate mean values (±SE). Dissimilar letters in a column represent significant variation (*p* = 0.05, LSD) and bold numbers are significant *p*-values.

**Table 3 ijms-24-06620-t003:** List of CFS factor treatments used during greenhouse experiments for potato (except for the negative control, the first number for each treatment indicates the pH of origin; names of the strains are coded by EB2004S and EL2006H for *B. subtilis* and *L. helveticus*, respectively; MRS and M13 stand for medium controls).

Experiment 1 Treatments	Experiment 2 Treatments
H_2_OCONT∼Negative control	H_2_OCONT∼Negative control
51EL2006H∼1%	72EB2004S∼0.4%
51MRSCONT∼1% (Positive control)	72M13CONT∼0.4% (Positive control)
52EL2006H∼0.4%	71EL2006H∼1%
52MRSCONT∼0.4% (Positive control)	71MRSCONT∼1% (Positive control)
52EB2004S∼0.4%	83EB2004S∼0.2%
52M13CONT∼0.4% (Positive control)	83M13CONT∼0.2% (Positive control)

## Data Availability

Data supporting reported results and conclusions will be made available by the authors, without undue reservation.

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
