# Peer review of "Cell-Free Supernatant (CFS) from Bacillus subtilis EB2004S and Lactobacillus helveticus EL2006H Cultured at a Range of pH Levels Modulates Potato Plant Growth under Greenhouse Conditions"

_ijms, 2023, doi:10.3390/ijms24076620_

Round 1
Reviewer 1 Report
The manuscript was very well written, except that the abstract is too long. I suggested the authors reduce the abstract (currently has more than 400 words), and then the manuscript can be considered appropriate to be published.
Author Response
Reviewer 1
We thank the reviewer for reading and giving suggestions to improve the manuscript.
The manuscript was very well written, except that the abstract is too long. I suggested the authors reduce the abstract (currently has more than 400 words), and then the manuscript can be considered appropriate to be published.
Response: The abstract is now shortened by approximately half of the original as shown here and in the revised manuscript: ‘Agriculture involving industrial fertilizers is another major human made contributing factor to soil pH variation after the natural factors like soil parent rock, weathering time span, climate, and vegetation. The current study assessed the potential effect of cell free supernatant (CFS) obtained from Bacillus subtilis EB2004S and Lactobacillus helveticus EL2006H cultured at three pH levels (5, 7, and 8) on potato (var Goldrush) growth enhancement under greenhouse pot experiment. The results showed that the main effect of CFSs obtained from B. subtilis EB2004S and L. helveticus EL2006H cultured at pH 5 significantly improved photosynthetic rates, stomatal conductance, root fresh weight and whole plant fresh weight. Interactive effects of pot pH and that of CFSs obtained from pH 5 influenced chlorophyll and plant height, shoot and whole plant fresh weight. Moreover, treatment 52EB2004S ⁓ 0.4% initiated early tuberization for potato grown at pH 7 and 8. Potato grown at pH 5 which received a 72EB2004S ⁓ 0.4% CFS treatment, caused greater whole plant fresh and dry weight than treatment with L. helveticus EL2006H CFS and a positive control. Taken together, the findings of this study are unique in that it probed the effect of CFS produced under differing pH conditions which revealed a new possibility to mitigate stresses in plants.’
Reviewer 2 Report
1- Please put the captions of the acronyms: CFs, EVL, DAP
2- In the introduction there should be a paragraph about the importance of growth promoting bacteria and the type of services they can do
3- In the experiments clearly state the type of substrate used for potato cultivation. I could only visualize the type of substrate at the end of the experiment description.
4- Item 2.1 is confused improve writing
5- In results, directly describe the results and do not rewrite the method.
6- Format all figures. There are figures without a horizontal axis and others of different sizes.
7- The information at the beginning of the discussion should be in the introduction.
8- Objectives is repeated on line 70.
9- On line 88, unnecessary information
10 - Is there an evaluation of the compounds contained in the CFs? Could you put?
11- You need to write in the material and methods why these strains were chosen.
12- Very extensive conclusion. This item should be short and direct.
Author Response
Reviewer 2
We thank the reviewer for the constructive suggestions for the improvement and clarity of the manuscript. The responses to each of the suggestions and comments are as elaborated below.
- Please put the captions of the acronyms: CFS, EVL, DAP
Response: The captions now inserted in the manuscript as follows: Cell free supernatant (CFS), EVL is a company name has no long form and days after planting (DAP).
- In the introduction there should be a paragraph about the importance of growth promoting bacteria and the type of services they can do
Response: A paragraph now added in the manuscript as follows: ‘The association of plants and microbes is being clearer in both positive and negative effects. Plant growth promoting microbes (PGPM) use various mechanisms for plant growth promotion which include bioactive compounds (phytohormones, peptides and volatile compounds) production, mineral solubilization, N2-fixation and enhanced nutrient use [27], and production of growth stimulating signal compounds [28]. Mechanisms related to antagonistic behavior and/or antibiosis also indirectly enhance plant growth and stress tolerance [28, 29]. Given the current understanding of these ecosystem services, the full force of the considerable potential benefits is yet to be determined. It is known that the efficacy of PGPM inoculation depends on soil pH, ability of the microbe to compete with the native strains, temperature, and host specificity [30].’
- In the experiments clearly state the type of substrate used for potato cultivation. I could only visualize the type of substrate at the end of the experiment description.
Response: Now the rooting medium used clearly stated in section 2.3: ‘Perlite was used in this experiment because it has greater has a neutral pH which corresponds to the reference pH of the treatments. Furthermore, perlite has high moisture retention, aeration and no nutrients which allowed equal supply of nutrient from the nutrient solution’.
- Item 2.1 is confused improve writing.
Response: Section 2.1 is now improved and reads in the manuscript as follows: ‘The microbial strains used in this study were obtained from EVL Inc.(http://www.evlbiotec.com), as a subset of a microbial biostimulant consortium in 2018. The two bacteria, B. subtilis EB2004S and L. helveticus EL2006H were selected based on previous reported effect of their CFSs on enhancement of seed germination and seedling growth for corn and tomato [40]. The strains were revived from working stock cultures stored at −20 ℃; a loopful was added into 50 mL fresh sterile M13 (B. subtilis EB2004S) or MRS (L. helveticus EL2006H ) medium broth, contained in 250 mL flasks. The cultures were then incubated for 48 hours at 30 and 37 ℃ for B. subtilis EB2004S and L. helveticus EL2006H respectively. An additional incubation condition included the use of an orbital shaker at 120 rpm was used for B. subtilis EB2004S, while L. helveticus EL2006H was incubated without agitation.
To obtain CFSs, strain cultures after 48 h incubation were transferred to sterile falcon tubes for cell separation by centrifugation at 12,000 g. The supernatant was filtered using vacuum filters (0.22 µm pore size) (Corning® Vacuum Filter systems, Fisher Scientific, Tewksbury, USA) to ensure no cells or debris remained. The obtained CFSs from pHs 5, 7 and 8 for B. subtilis EB2004S and pHs 5 and 7 for L. helveticus EL2006H, including sterile medium (positive control) at respective pHs and a negative control were used for treatments as indicated in Table 1. For final treatment formulations a ratio of CFS to distilled water (volume of CFS/volume of distilled water) was used to get 1, 0.4 and 0.2 % (Table 1)’.
- In results, directly describe the results and do not rewrite the method.
Response: Sentences indicating method in the results section are now removed as indicated in the revised manuscript.
- Format all figures. There are figures without a horizontal axis and others of different sizes.
Response: All figures are formatted as suggested and are now replaced as indicated in the revised manuscript.
- The information at the beginning of the discussion should be in the introduction.
Response: Part of information at the beginning of discussion is now moved to introduction section as indicated in the revised manuscript.
- Objectives is repeated on line 70.
Response: Repetition of the objectives is now omitted in the manuscript.
- On line 88, unnecessary information
Response: The line is now deleted from as indicated in a revised manuscript.
- Is there an evaluation of the compounds contained in the CFs? Could you put?
Response: Unfortunately, in the current study CFS compounds were not evaluated, we thank the reviewer for this constructive suggestion to further study on the compounds responsible for the beneficial effects on potato plant growth.
- You need to write in the material and methods why these strains were chosen.
Response: The reason for selection of strains is now added in section 2.1.
- Very extensive conclusion. This item should be short and direct.
Response
The conclusion is now shortened as indicated in the revised manuscript.